# Effect of Laser Energy Density, Internal Porosity and Heat Treatment on Mechanical Behavior of Biomedical Ti6Al4V Alloy Obtained with DMLS Technology

**DOI:** 10.3390/ma12142331

**Published:** 2019-07-22

**Authors:** Mierzejewska Żaneta Anna

**Affiliations:** Faculty of Mechanical Engineering, Bialystok University of Technology, Wiejska 45c Street, 15-351 Białystok, Poland; a.mierzejewska@pb.edu.pl; Tel.: +48-692-885-870

**Keywords:** Selective Laser Melting, Direct Metal Laser Sintering, porosity, titanium alloys, yield strength, ultimate tensile strength, X-Ray Diffraction

## Abstract

The purpose of this paper was to determine the influence of selected parameters of Direct Metal Laser Sintering and various heat treatment temperatures on the mechanical properties of Ti6Al4V samples oriented vertically (V, ZX) and horizontally (H, XZ). The performed micro-CT scans of as-build samples revealed that the change in laser energy density significantly influences the change in porosity of the material, which the parameters (130–210 W; 300–1300 mm/s), from 9.31% (130 W, 1300 mm/s) to 0.16% (190 W, 500 mm/s) are given. The mechanical properties, ultimate tensile strength (UTS, Rm) and yield strength (YS, Re) of the DMLS as-build samples, were higher than the ASTM F 1472 standard suggestion (UTS = 1100.13 ± 126.17 MPa, YS = 1065.46 ± 127.91 MPa), and simultaneously, the elongation at break was lower than required for biomedical implants (A = 4.23 ± 1.24%). The low ductility and high UTS were caused by a specific microstructure made of α’ martensite and columnar prior β grains. X-Ray Diffraction (XRD) analysis revealed that heat treatment at 850 °C for 2 h caused the change of the microstructure intothe α + β combination, affecting the change of strength parameters—a reduction of UTS and YS with the simultaneous increase in elongation (A). Thus, properties similar to those indicated by the standard were obtained (UTS = 908.63 ± 119.49 MPa, YS = 795.9 ± 159.32 MPa, A = 8.72 ± 2.51%), while the porosity remained almost unchanged. Moreover, the heat treatment at 850 °C resulted in the disappearance of anisotropic material properties caused by the layered structure (UTS_ZX_ = 908.36 ± 122.79 MPa, UTS_XZ_ = 908.97 ± 118.198 MPa, YS_ZX_ = 807.83 ± 124.05 MPa, YS_XZ_ = 810.56 ± 124.05 MPa, A_ZX_ = 8.75 ± 2.65%, and A_XZ_ = 8.68 ± 2.41%).

## 1. Introduction

Direct Metal Laser Sintering (DMLS) is predicted to revolutionize the implants manufacturing. However, it is crucial to understand the possibilities and limitations of the process for the Ti6Al4V alloy, which is still the subject of extensive research. Phase transitions, as a result of heat treatment, have a considerable influence on the mechanical properties of the material [1,2], as well as α lath thickness [3]. Hrabe and Quinn [4] conducted a series of experiments to determine the effect of a microstructure on the properties of the samples in order to compare the mechanical properties of the Ti6Al4V alloy. Their work revealed that the orientation of the base plane in relation to the load direction is of great importance. The yield strength was the highest when the direction of the load was perpendicular to the base plane and the lowest in the orientation of an angle of 45°, resulting in a maximum shear stress at the basic plane. They have also proved that the texture has almost no effect on the ductility.

In numerous research papers, it has been proved that the Ti6Al4V alloy’s microstructure created with a laser beam has some unique features that are characteristic of DMLS [5]. Tests of Ti6Al4V samples made with the use of Nd: YAG lasers presented that columnar grains are a common feature of microstructures made with additive techniques [6,7,8,9]. These grains grow along the boundaries of the original β grains, along with the build direction, as a result of repeated cycles above the β-transus and solidification temperatures, which start at a temperature of about 1660 °C. When passing through the liquidus line, the grain growth of the primary β phase occurs very quickly, making the grains orientation vertical, according to the build direction [10,11]. The laser beam creates a pool of molten material, from which the heat is transferred to the environment and to the already solidified layers. As the material below the top layer is again partially melted, the grains grow epitaxially from the layer below [12,13,14]. Some researchers suggested that the presence of columnar grains in DMLS samples may result in anisotropic mechanical properties of as-build samples (immediately after sintering) [15,16]. Regarding mechanical properties, it has been shown that cracks tend to spread along columnar grains [17,18]. Depending on the direction of tensile forces and the orientation of the columnar grains, there were differences in the ductility of samples built in a vertical or horizontal direction [7,19,20]. While the tensile strength was approx. 1060 MPa in both directions, the elongation was 11% and 14% for samples build in vertical and horizontal orientation, respectively.

The microstructure of the Ti6Al4V alloy produced by SLM technology is entirely made of the martensitic α’phase [10,21]. This is a result of high rates of heating and cooling of the solidified powder, which are neglected in the DMLS process [22]. Fine martensitic structure α’determines a high yield point and ultimate tensile strength (UTS) of more than 1 GPa, but also has a low elongation value, not exceeding 10%, and therefore lower than forged and casted materials [6,7,10,23]. Vranckenet al. analyzed the effect of various heat treatments on the microstructure of samples manufactured with SLM technology, taking into account different times, temperatures, and cooling rates [24]. The original α’structure has been transformed into a α + β plate mixture with heating below β-transus, while the features of the original microstructure (columnar grains) were preserved. After the heat treatment above the β-transus temperature, a clear change in the microstructure was observed—the columnar β grains were transformed into equiaxed grains.

The presence of defects in the microstructure, which are the source of initiation and propagation of cracks, as well as their random distribution, influences the reduction of fracture strength and fatigue strength due to significant stress peaks [25,26,27,28]. The morphology, number, size, and location of defects also affect the durability of samples created with the use of SLM technology. Qui et al., Liu et al., Gong et al., and Zhang et al. proved that spherical defects had less impact on durability if their volume did not exceed 1% [7,26,29,30]. The fracture toughness was significantly lowered, while spherical defects created over 5% of material porosity. Irregular defects in the volume of up to 1% turned out to be much more harmful for the samples tested in terms of mechanical strength. Kasperovich et al. tested samples that were subjected to various types of post-treatment—their aim was crack elimination, which occur in the structure, to improve fatigue strength [31]. Their research indicated that the heat treatment process only affected the change of the microstructure, however, it did not eliminate its defects, and therefore the improvement of fatigue life was negligible. As it is difficult to control the type, quantity, and location of defects in DMLS parts, fatigue strength can be different, even when the manufacturing process is carried out using regular parameters. Therefore, the fatigue strength of components manufactured by SLM technology still raises doubts and requires further research to be improved.

A number of studies conducted by Leuders et al., Qui et al., and Simonelli et al. have shown that it is possible to obtain a density of more than 99.5% for Ti6Al4V samples produced with the use of a laser beam [6,7,10]. Some researchers suggest that, due to the apparent density of the powder (50–60%), small amounts of gas may be entrapped in the material after solidification, resulting in the pores formation, which can be defined as small and empty spaces inside the solid material [3,32]. Gas pores are detrimental to mechanical properties because they act as stress concentrators in components under load, reducing fatigue life and elongation [7,25,33,34]. For this reason, one of the objectives of this work was to analyze the effect of porosity on the mechanical properties of samples obtained with DMLS technology.

The unique properties of titanium in medical applications were discovered relatively recently in the mid-1970s.Pioneering research, which was conducted in 1983, presented that this material promotes bone build-up on its surface (contact osteogenesis), causing a slight inflammatory reaction of tissues to foreign bodies [35]. One of the alloys meeting the requirements for biomaterials is a two-phase alloy Ti6Al4V (Titanium Grade 5/UNS R56400/WNR 3.7165), consisting of aluminum (6%) and vanadium (4%). This alloy is used in many industries, but mainly in biomedical engineering [36].

Concern about the validity of the titanium alloy in biomedical applications arouse the presence of aluminum and vanadium, and its toxicity to living organisms have been well documented [37]. However, specialists in the field of materials and medicine ensure that the occurrence of complications associated with the release of corrosion products to the tissues surrounding the implant is negligible, as titanium and its alloys undergo spontaneous surface passivation [38]. This process is related to the spontaneous formation of a tight and stable TiO_2_ layer, formed on the surface, protecting the implant from the aggressive tissue environment and corrosion processes [39]. The unique properties of the Ti6Al4V alloy such as high corrosion resistance, so-called relative strength, low modulus of elasticity, high biocompatibility, excellent osseointegration, or low density, allow for the comprehensive use of its potential [40]. On the basis of ASTM F 1472 specifications for a forged Ti6Al4V alloy used as a medical material, the YS and UTS should not be lower than YS = 868 MPa and UTS = 930 MPa, respectively, and the elongation at the moment of break should be no less than 10%. Obtaining such properties with additive technologies requires a precise selection of DMLS parameters and additional heat treatment, due to which the stresses in the material will be eliminated, and the single-phase martensitic structure will decompose into a two-phase structure suitable for implantable biomaterials.

While the technology of selective laser melting is standardized, each commercial DMLS system presented in the published studies shows a separate characteristic in terms of materials and basic process characteristics (powder delivery to the working field, platform temperature, optical system, and laser) that affect the obtained results [41,42]. Literature analysis indicates a lack of a systematic approach to the study of laser–metal interactions, which would be beneficial for the further development of DMLS. To determine the optimal process parameters, extensive and comprehensive research on the relation of process conditions and product properties are required. Therefore, the key to this study was to investigate how the orientation affects the strength of the material and whether the heat treatment temperature can affect the reduction of the anisotropy of the material obtained with DMLS technology

## 2. Materials and Methods

The verification of the chemical composition of the analyzed powder was carried out using Thermo ARL Quantris spectrometer Spectrometer (Thermo Fisher Scientific, Waltham, MA, USA). Detailed results of the powder analysis are presented in Table 1. The content of Al and V was within the range specified in ASTMF 2924-14 (Standard Specification for Additive Manufacturing Titanium-6 Aluminum-4 Vanadium with Powder Bed Fusion).

Powder particle size measurements were performed using the ANALYSETTE 22 Micro Tec plus particle analyzer from FRITSCH GmbH Milling and Sizing (Idar-Oberstein, Germany). According to the powder producer (EOS company, Krailling, Germany), the average particle size is around 30 μm. However, laboratory analyzes revealed that the particle size was in the range of 10 to 100 μm. The average particle size obtained for three subsequent measurements of the particle distribution was equal to 39.81 μm, with about 94.2% of the particles investigated ranging from 20 to 80 μm (0–30 μm, 18.12%; 30–80 μm, 80.82%, 80–100 μm, and 1.10%). The grains observed under the microscope were spherical and smooth, with numerous satellites, which correspond to the morphology of gas sprayed powder. The advantage of spherical morphology and wide particle size distribution has high fluidity and high packing density.

Thirty groups of test samples were used in experimental research. In each group, different laser powers (130 W, 150 W, 170 W, 190 W, and 210 W) and different beam speeds (300 mm/s, 500 mm/s, 700 mm/s, 900 mm/s, 1100 mm/s, and 1300 mm/s) were used. In all samples, a cross–scan pattern was used and the same distance of scanning vectors was maintained, as well as layer thickness (0.03 mm), spot size (0.1 mm), and hatch distance (0.1 mm). The process parameters used in the present study have been presented in Table 2.

The energy density (E) was calculated based on laser power (P), the distance between the laser scan line (h), scanning speed (v), and layer thickness (t), which is presented in Equation (1) [13].

(1)$$E\;=\;\frac{P}{\mathrm{vht}}\;[J/{\mathrm{mm}}^{3}]$$

DMLS samples were divided into five groups—four of them were heat treated and one remained in the as-build state. Samples were heated for 2 h at 650 °C, 750 °C, 850 °C, and 950 °C. Heat treatment was carried out in a vertical furnace, heating at a rate of about 10 °C/ min, under argon atmosphere (to prevent oxidation of the titanium surface). The samples were cooled with the furnace to 500 °C at a rate of about 0.04 °C/s, and then cooled in the air. After heat treatment, the samples were grinded with abrasive paper with gradations of 200, 500, and 800, purified in 70% ethanol, rinsed with deionized water, and then electropolished in a bath solution using a perchloric acid, glacial acetic acid, and distilled water mixture in a volume ratio of 1:10:1.2 [43]. The current density was equal to 0.3 A/cm^2^. The samples were treated with fresh solution at room temperature (25 °C) for 15 min. The Bruker Sky Scan 1172 (Brucker, Billerica, MA, USA) scanner with set parameters: Number of rows—2664, number of columns—4000, Au + CU filter, sample rotation—0.2500, and pixel size—4.28 μm was used for microtomographic optical analysis. Scanned images of samples were reconstructed with the use of software dedicated to micro-CT image analysis—NRecon, Data Viewer, CTvox, and CTAn. Tensile properties were studied using a Hegewald and Peschke INSPEKT (Meß—und Prüftechnik GmbH, Nossen, Germany) test machine with a maximum breaking strength of 5 kN. Displacements were measured by using an extensometer with a 25 mm gauge length. Yield stress and Young’s modulus were determined according to ASTM E 111. XRD X–ray phase analysis was performed using a Bruker D8 Advance Eco-ray diffractometer (Brucker, Billerica, MA, USA), equipped with a Cu Kα1 X-ray tube = 1.5406 Å and an SSD160 detector (Brucker, Billerica, MA, USA) adapted for ultra–fast XRD diffraction detection.

## 3. Results and Discussion

All samples were produced on one mounting plate. Due to the specificity of the process, changes in the dimensional accuracy of the samples with respect to nominal dimensions (Figure 1a) were expected. The dimensional accuracy was affected by the intensive degreasing of samples in an ultrasonic washer and resulted in the rinsing out of loose, unmelted powder grains, and consequently, changes in the dimensions of the samples. Thirty randomly selected samples were subjected to detailed measurements in strictly defined places—fourteen points, marked A through M, presented in Figure 1b.

The dimensions of each sample were measured with a digital caliper to an accuracy of 0.01 mm. The list of measurements is presented in Table 3.

### 3.1. Porosity

Polished and etched surfaces of cylindrical samples were subjected to microscopic observations. The effect can be seen in Figure 2. It was concluded that the change in scanning speed as well as the change in laser power significantly affected the shape and the number of pores.

The quantitative analyzes allowed for a precise assessment of the samples porosity by means of non-destructive microtomographic examination. The correlation between the decreasing value of the melting energy density and the increasing porosity is highly noticeable (Figure 3). However, porosity increases significantly with the increase of scanning speed. This regularity was observed in each group of samples. For samples melted with a laser power of 130 W and a scanning speed of 500 mm/s, the material density was almost 99%. The porosity in the range of 0.72% indicates the presence of internal defects in the structure. An increase is observed below and above 500 mm/s in porosity—from 1.27% at a speed of 300 mm/s to 9.31% at a speed of 1300 mm/s (Figure 3a). Samples melted with a power of 150 W and 300 mm/s were characterized by a porosity of 1.5%. Above 500 mm/s, an increase in porosity was observed—from 3.37% at a speed of 700 mm/s to 8.43% at a speed of 1300 mm/s. The minimum value porosity of 0.46% was observed at a laser speed of 500 mm/s. In comparison to samples manufactured at 130 W with the same speed, the porosity was lower by 0.26%. The difference between the maximum porosity values obtained for the highest laser speed was less than 1% (Figure 3b). For samples melted with a laser power of 170 W at scanning speeds from 300 to 700 mm/s, the porosity of the material varies from 1.69% to 0.8%. Above 700 mm/s, an increase in porosity is observed—from 5.29% at a speed of 900 mm/s to 7.29% at a speed of 1300 mm/s. Samples melted with a power of 170 W achieved a density of more than 99% for two scanning speeds—500 and 700 mm/s. The highest density, 99.73%, was obtained for samples scanned with 700 mm/s (Figure 3c). Additionally, for samples melted with a laser power of 190 W and scanning speed of 500 mm/s and 700 mm/s, porosity did not exceed 1%. The lowest value of porosity was observed for samples melted with 500 mm/s—0.16%. At a scanning speed of 300 mm/s, microstructure defects in the analyzed samples were slightly below 2%, while a further increase in scanning speed above 500 mm/s resulted in higher porosity—from 0.68% for 700 mm/s to 6.57% for 1300 mm/s (Figure 3d). Similar characteristics were found for samples melted with a power of 210 W. The porosity obtained at 300 mm/s exceeded 2% and a decreasing tendency was observed along with the increase of sintering speed to 700 mm/s. After exceeding 700 mm/s, the porosity increased again, reaching a maximum value of 6.16% at a sintering speed of 1300 mm/s. The lowest porosity was obtained at 700 mm/s and was 0.45% (Figure 3e).

At low energy density (33–71 J/mm^3^), the porosity ranges from 9.31% to 3.37%. The pores are unevenly distributed, irregular in shape, and interconnected. Porosity is characterized by large recesses filled with loose particles of unmelted grains. A possible explanation for this is a low energy density and relatively small depth of laser penetration, which make the size of the melt pool too small as the powder particles are not sufficiently liquefied to completely melt, but provides a sufficient bond between the layers. The increase in energy density from 78 J/mm^3^ to 127 J/mm^3^ generates a relatively high temperature, which facilitates the flow of liquid and fills the space between already melted grains, causing the porosity in the range of 0.84–0.16%.

At higher energy densities, the pores are small and mostly spherical, and their formation is most often associated with gas bubbles trapped under the molten powder layer. An increase in the laser energy density above 127 J/mm^3^ causes changes in the pore morphology and an increase in porosity from 1.12% for 140 J/mm^3^ to 2.18% for 233 J/mm^3^ (Figure 4).

The influence of energy density on porosity was also investigated by Han et al. [44], Laquai et al. [45], and Dilip et al. [46]. While the range of the optimal power density indicated by Han and Lagui is similar to that obtained in the present study (120–180 J/mm^3^ and 120–195 J/mm^3^, respectively), significant differences were observed. Both authors state that the porosity in the tested samples did not exceed 0.05%. What is more, Han also revealed that, in the range of much lower (60 J/mm^3^) and much higher (240 J/mm^3^) energy density, the density of melted material is not less than 99.75%. Lagui, on the other hand, observed an increase in porosity just above 1% for only energy densities below 50 J/mm^3^ and above 300 J/mm^3^. Completely different characteristics were presented by Dilip et al. [47]. They showed that the optimal range of energy density was between 50 and 66 J/mm^3^ (porosity below 0.5%), for energy densities around 40 J/mm^3^, the porosity increased to about 5%, while at higher energy densities between 90 and 130 J/mm^3^, the porosity was about 10%.

### 3.2. Mechanical Properties

The layered structure obtained in DMLS technology makes the samples characterized by anisotropic mechanical properties. For this purpose, two types of samples were printed—vertically oriented samples (V, ZX) and horizontally orientated samples (H, XZ; Figure 5). The layering in both samples was perpendicular to the building direction (Z), while the tensile forces acting on the samples were oriented perpendicular to the layers in the ZX samples and parallel to the layers in the XZ samples. The tensile strength results presented in Figure 6, Figure 7, Figure 8, Figure 9 and Figure 10 were calculated on the basis of five consecutive tests.

#### 3.2.1. As-build Samples

Based on ASTM F 1472 specifications for the forged Ti6Al4V alloy used as medical materials, YS and UTS should not be lower than 868 and 930 MPa, respectively, and elongation at break should be no less than 10%. For all as-built samples (Figure 6a–c), the yield strength and tensile strength significantly exceeded the minimum requirements for the forged material described in the standard and were equal to 1065.46 ± 127.91 MPa and 1100.13 ± 126.17 MPa, respectively. The highest strength was characterized by samples melted with 190 W power and 500 mm/s speed (Re max = 1261 ± 5.65 MPa and Rm max = 1288 ± 28.28 MPa), while the lowest was the samples melted at 130 W and a speed of 1300 mm/s (Re min = 885 ± 7.07 MPa; Rm min = 918.5 ± 28.99 MPa). In the group of samples melted with 130 W and 1100–1300 mm/s, strength parameters did not meet the criteria of ASTM F 472.The fragility of the martensitic microstructure and the presence of residual stresses are responsible for the lower ductility.

It can be determined that the samples elongation was definitely lower than required in this standard, and its average value was 4.23 ± 1.24%, with the maximum elongation in this group of samples being A max = 5.8 ± 0.84%, and the minimum A min = 2.2 ± 0.56%. The tensile strength of Ti6Al4V samples produced by the DMLS method is only slightly higher than their yield strength, which indicates the brittleness of the material and its low ductility. An increase scanning speed above 700 mm/s in almost every sample group influences the reduction of the strength parameters and elongation. This is due to the fact that higher speeds cause the generation of a lower energy density, which results in structural defects and the weakening of bonds between layers. Additionally, the increase in power affected the changes of mechanical parameters—tensile strength increased from 1050 ± 136.97 MPa (130 W) to 1148.83 ± 106.74 MPa (210 W). YS and A also increased—from 1013.58 ± 134.95 MPa to 1112.41 ± 109.85 MPa and from 3.72 ± 1.43% to 4.72 ± 0.99%.

The surfaces fracture of non-heat treated samples showed mixed characteristic of brittle and ductile fracture, accompanied by very small plastic deformation. Fracture surfaces were characterized by the presence of shallow dimples on quasi cleavage surface, delamination, or stepped cracks, suggesting intergranular destruction along hard, prone to cracking, brittle needles α’, which then propagated due to decohesion caused by further deformation of the sample. A large number of gas defects and pores caused the concentration of stress and violent crack propagation, and the mixed nature of cracking to a limited elongation (Figure 6d).

#### 3.2.2. Heat Treatment at 650 °C

Despite heat treatment, the results of tensile and yield strength were still higher than the minimum determined by the ASTM F 1472 standard (Rm = 1071.7 ± 128.08 MPa; Re = 994.85 ± 124.91 MPa), and the elongation was also unsatisfactory and almost a half lower than required (A = 4.24 ± 1.25%). The highest strength was characterized by samples melted with the power of 190 W and the speed of 500 mm/s (Rm max = 1237 ± 26.87 MPa; Re max = 1192.5 ± 6.36 MPa), the lowest was the samples melted with the power of 130 W at a speed of 1300 mm/s (Rm min = 864.5 ± 24.74 MPa; Re min = 810 MPa). The tensile strength of ZX samples decreased to an average of 1050.73 ± 145.97 MPa, and XZ samples to 1023.86 ± 206.64 MPa. With the decrease in strength properties, the elongation increased for ZX samples to 4.27 ± 1.57%, and for XZ samples to 4.21 ± 0.83%. The differences in tensile strength between the samples printed in the XZ plane and ZX plane was approx. 3.3% (Figure 7a–c). Obtained results clearly indicate that annealed samples had higher elongation at break—regardless of their orientation, and thus better ductility, in comparison to as-build samples. A large range of values from average elongation at break indicates that the improvement is not as significant as in other works presented in the literature [24,25,45]. The minimum elongation value, according to the standard, cannot be lower than 10%, which in the case of the applied heat treatment, was not obtained for any of the samples, therefore reducing the stresses turned out to be beneficial for DMLS Ti6Al4V, but is still insufficient to meet the criteria imposed by ASTM F 1472. The surface of the breakthrough remained mixed, with features of brittle and ductile fractures, with shallow dimples staircase cracks and pores (Figure 7d).

#### 3.2.3. Heat Treatment at 750 °C

Annealing at 750 °C for 2 h significantly reduced the strength parameters and increased the plasticity of the material, as a result of partial phase transformation α’ → α + β and at the same time caused the growth of grain α, thus changing the phase composition and grain morphology (in relation to as-build samples).

For heat treatment at 750 °C, the Rm of the samples was equal to 1002.88 ± 119.42 MPa, while Re = 900.07 ± 77.43 MPa. The highest strength was again achieved in samples melted with a power of 190 W and a speed of 500 mm/s (Rm max = 1169.5 ± 6.36 MPa; Re max = 1085.5 ± 13.43 MPa). The lowest tensile strength and yield strength were obtained in samples melted with a power of 130 W and a speed of 1300 mm/s again—the average value of Rm min = 812 ± 11.31 MPa; Re min = 723 ± 12.72 MPa. Heat treatment also had a slight influence on the elongation—increased from A = 5.24 ± 1.25% to A = 5.71 ± 1.69%. The difference in the strength of the ZX and XZ samples also decreased: ZX Rm = 1002.27 ± 126.06 MPa, XZ Rm = 998.72 ± 113.82 MPa, similar to the yield strength—Re = 899.53 ± 133.26 MPa (for ZX), Re = 900.6 ± 115.76 MPa (for XZ). The elongation for ZX samples increased to 5.75 ± 2.07%, while for XZ samples, to 5.67 ± 1.23% (Figure 8a–c). The nature of the fractures area has changed slightly, showing more features of a ductile fracture than in the case of previous samples. On the fracture surfaces, defects of the microstructure, i.e., pores, cracks, and discontinuities of the material, were visible (Figure 8d).

#### 3.2.4. Heat Treatment at 850 °C

Heat treatment at 850 °C led to a change in the microstructure and decomposition of the α’phase to the α + β. Therefore, it can be assumed that the samples annealed at this temperature presented the best relationship between strength and elongation. The average tensile strength value for all analyzed samples was equal to 908.63 ± 119.49 MPa, yield strength 795.9 ± 159.32 MPa, and elongation at break 8.72 ± 2.51%. The highest tensile and yield strength were again characterized by samples of 190 W/500 mm/s (Rm max = 1058 ± 4.94 MPa; Re max = 958 ± 4.95 MPa), the lowest was again the samples of 130 W/1300 mm/s. The average value of Rm min was 724.5 ± 4.95 MPa, and Re min = 625.5 ± 6.36 MPa. A significant part of the samples treated at this temperature does not meet the criteria of ASTM F 1472 in terms of tensile strength and yield strength, but they still meet the criteria of ASTM F 1108-14.

What is more, processing at 850 °C resulted in homogenization of the microstructure, and as a result, anisotropic properties of differently orientated samples have disappeared. The tensile strength of ZX samples was therefore Rm = 908.36 ± 122.79 MPa, samples XZ Rm = 908.97 ± 118.198 MPa, the yield strength Re for both orientations was equal to: 807.83 ± 124.05 MPa and 810.56 ± 124.05 MPa, and elongation A: 8.75 ± 2.65% and 8.68 ± 2.41% (Figure 9a–c). The surface fracture is typically ductile, with dimples and voids, in the middle of which there are particles of inclusions. In the central part of the crack, the dimples are equi-axial, while on the sides they have a strong shearing character and a parabolic shape (Figure 9d).

#### 3.2.5. Heat Treatment at 950 °C

Heat treatment at 950 °C significantly affected the change in the microstructure—similarly to that in the lower temperature treatment, the α’martensitic phase decomposed into the α + β phase mixture, but temperature close to β-trasus caused increased grain size and deterioration of plastic properties. Therefore, along with the decrease in the value of strength properties, the elongation at break decreases. As a result, the analyzed properties have the following values: Tensile strength Rm for all analyzed samples is 848.48 ± 86.56 MPa, yield strength 781.35 ± 96.3 MPa, and elongation at break 7.75 ± 2.69%. The highest measured values of tensile strength and yield stress are, respectively, Rm max = 931 ± 1.41 MPa; Re max = 927.5 ± 3.54 MPa, while the lowest—Rm min = 702.5 ± 12.5 MPa and Re min = 615.5 ± 6.36 MPa (Figure 10a–c). The character of the fracture surface remains ductile (Figure 10d).

The obtained data is supplemented by the assessment of the influence of porosity on the strength and elongation of samples, the yield point and Young’s modulus (Figure 11 and Figure 12). With the increase of porosity, all these parameters revealed a downward trend, which clearly indicates that defects of the microstructure strongly affect the mechanical properties of the material.

The analysis of the obtained results revealed that samples build in the ZX orientation compared with XZ had a higher yield strength, tensile strength, and elongation, if the volume of the microstructure defects does not exceed 2%. Vertical samples (ZX)—due to the layered structure—are seemingly more prone to initiation and cracks propagation than horizontal samples (XZ), however, they showed higher tensile strength and yield strength. The explanation of this phenomenon is related to the microstructure, especially to the columnar β grains. These grains grown perpendicular to successive layers of molten powder and in accordance with the direction of building the element. As a result, in the XZ samples, tensile force direction was perpendicular to the β grains, whereas in the ZX samples, the direction of the force was parallel. In vertical samples, the grains were axially extended, which had a positive effect on the strength properties of the whole sample. The same orientation between the columnar grains and the direction of the tensile forces led to a reduction of the grain boundary resistance, which no longer constituted an effective obstacle to the dislocation movement and which could be subject to greater distortion [48,49]. Reducing the energy density by reducing the laser beam power and increasing the scanning speed resulted in defects between successive layers of material [50,51]. For highly porous samples, force was applied parallel to the layers and caused pores deformation consistent with the direction of force, whereby the stress concentration around the pores was several times lower, and consequently, the material was less susceptible to the initiation and propagation of cracks.

Based on the microstructure and changes in mechanical properties analysis, presented in the paper, DMLS parameters, application of which guarantees the achievement of a biomaterial with properties consistent with ASTM F 1142, were indicated in Table 4.

### 3.3. Microstructure Characterisation

Unique thermal features, such as heat input, large temperature gradients, and high cooling ratesdramatically affected as-built martensitic α’microstructures and led to high residual stresses. Annealing at 650 °C for 2 h was intended to remove residual stress from the material. The use of this heat treatment resulted in the decomposition of a small amount of the martensitic phase in the α + β equilibrium phase, as well as the nucleation and slow increase of the α phase at the boundaries of martensitic needles. Nevertheless, the microstructure has not changed significantly compared to untreated, and the columnar grain structure has been retained. This is due to the fact that rapid cooling during the DMLS process leads to the formation of dislocation within the alloy. These defects have an inhibitory effect on the α-phase distribution. Furthermore, during annealing at 650 °C, which is significantly below than β–transus, the driving force of the phase distribution is insufficient. The microstructure after heat treatment at 650 °C still remained needle-shaped (Figure 13a). The only noticeable change was the increase in grain thickness to 0.6 μm (Table 5). 

The microstructure of samples annealed at 750 °C for 2 h and cooled together with the furnace clearly indicate the decomposition of fine martensite into α and β mixture, in which the α phase occurs in the form of fine needles (Figure 13b). The analyzed microstructure consisted of a smaller amount of α’ martensite embedded in the more stable α and β phase. According to the findings of Shunmugavel et al., the upper limit of martensite decay in the Ti6Al4V alloy is 800 °C, which means that only above this temperature martensite is completely decomposed into the phases α and β [14]. The average thickness of α’ plates was 1.2 μm.

Heating of samples at 850 °C for 2 h significantly influenced the changes of the microstructure—the α’ phase completely decomposed into the mixture of α and β phases. The α grain thickness increased to the average value 1.5 μm. Heat treatment carried out at 850 °C did not change the morphology of the primary β grains. The reason for this is the coexistence of the α + β phases, which mutually inhibit their further growth. Different orientation of the growth of adjoining grains prevents the migration of boundaries in the axial direction, which leads to slight changes in their shape (Figure 13c).

The increase of the annealing temperature to 950 °C resulted in the reduction of the inhibitory effect of the grain boundaries—the structure was transformed into coarse grains. The average width of the plate grains of the α-phase after annealing was approximately 2.3 μm. This was the result of the increase in the driving force of the α → β phase decay and the joining of the adjacent β grains under the influence of high temperature. In the fully lamellar microstructure, the size of the α colony and the alternately packed α and β plates with a clear orientation define the grain and have a strong influence on the size and length of the slip, determining the ductility of the Ti6Al4V alloy (Figure 13d). Thus, after a heat treatment at 950 °C, a lower strain was expected during the static tensile test.

The analysis of the microstructure made of Ti6Al4V alloys produced under heat treatment was conducted by phase analysis (Figure 14). XRD spectra clearly presented that the α’ phase was the dominant fraction in as-build samples. All peaks untreated material were indexed as the α’ phase, while the β phase was not detected. Both martensitic phases α and α’ have similar crystallographic cell parameters. However, since fast cooling may promote martensitic transformation, it was expected that the hexagonal hcp phase in the material was acicular martensite. Therefore, metastable martensite remained as the only phase present at room temperature in the alloy Ti6Al4V produced by DMLS technology. In the material heat treated at 650 °C, the main peaks remained the same as in the as-build, however peaks (110) β-Ti2θ = 38.57° and (211) β-Ti (2θ = 71.02°) corresponded to the β phase. A similar pattern was obtained for heat treatment at 750 °C, only the intensity of α–peaks increased due to the increase in grain thickness. Figure 9 presented phase analysis for V oriented samples melted with an energy density of 100 J/mm^3^.

Heat treatment at 850 °C resulted in complete decomposition of α’ martensite into α + β phase and this influenced the XRD pattern obtained in the study, because peaks 2θ = 35.40°, 39.37°, 40.47°, 53.31°, and 63.54° corresponded to the plate α-phase, not the α’ phase. The peaks of this phase showed strong intensity, which was related to the increase in grain size. The intensity of peaks (110) β-Ti (2θ = 38.57°) and (211) β-Ti (2θ = 71.02°) also increased, which clearly indicates that a significant amount of β phase increased due to the redistribution of alloying elements. The XRD of the annealed material at 950 °C showed that the microstructure of the DMLS alloy formed a mixture of α + β phases. This unambiguously confirms that the transformation temperature of β–transus is higher than 950 °C.

The change of microstructure as a result of heat treatment affects not only mechanical properties, but also corrosion resistance. According to the literature, as-build samples show a poorer corrosion resistance compared to heat treated samples. Based on published studies, it can be concluded that the corrosion resistance depends on the volume of β phase—the more that is in the structure, the higher the corrosion resistance [52,53].

## 4. Conclusions

This work aimed to determine the influence of DMLS parameters on the resulting properties of the obtained material and to assess the effect of different heat treatment temperatures on the structure and mechanical properties ofsamples oriented vertically and horizontally. The results of completed research are the following conclusions:The results of porosity in samples made with DMLS technology differs from those presented in the literature; however, it should be noted that the micro-CT porosity analysis, which was carried out in this work, allowed the assessment of total porosity, whereas in studies presented in the data of the reference, these measurements were made on the basis of fragmentary image analysis or the Archimedes method, which allows the estimation of apparent density.At low energy density (33–71 J/mm^3^), the porosity varies from 9.31% to 3.37%; the increase in energy density from 78 J/mm^3^ to 127 J/mm^3^ causing the porosity in the range of 0.84–0.16%; above 127 J/mm^3^, the porosity increase again as the effect of the material overheating.The microstructure created in the DMLS process is fine-grained α’ martensite, which determines high strength properties and low ductility; this microstructure—due to the ferromagnetic properties of martensite—should not be considered as a biomaterial and requires operations whose aim is to change the microstructure to a biphasic phase and to obtain parameters compliant with the normative ones.The tests showed that the most favorable combination of mechanical properties and structure without the α’ phase may be obtained by annealing the material in the temperature range 850–950 °C for 2 h, cooled together with the furnace.After heat treatment, the samples showed YS and UTS similar to those for wrought and annealed Ti6Al4V—this was the effect of the coexistence of the α and β phases.Tensile strength tests showed the sensitivity of the as-build material for porosity and orientation of samples, which results from the creation of columnar grains growing along the boundaries of the prior β grains.Heat treatment influenced the reduction of tensile (Rm) and yield strength (Re) parameters with a simultaneous increase of elongation (A) and Young’s modulus (E), additionally, heat treatment at 850 °C resulted in homogenization of the microstructure and elimination of anisotropy resulting from different directions of stretching samples and layering.The highest density of samples (porosity not exceeding 0.5%) was obtained for samples melted with energy density in the range of 100–127 J/mm^3^.The analysis of the obtained results revealed that samples build in the ZX orientation compared with XZ had a higher yield strength, tensile strength, and elongation, if the volume of microstructure defects does not exceed 2%.Regardless of the energy density, the microstructure obtained in the DMLS process consists of martensite needles, however, the higher the energy density, the larger the grain size; the size of grains also grows with the increasing temperature of heat treatment, which affects the reduction of strength properties and increase of elongation.

## Figures and Tables

**Figure 1 materials-12-02331-f001:**
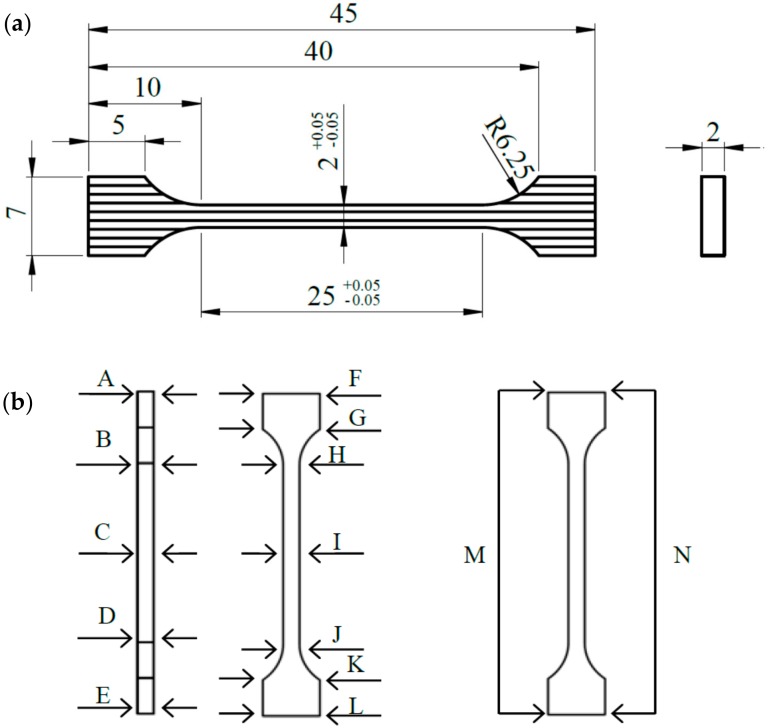
Nominal dimensions of a single sample (**a**) and measurement points (**b**).

**Figure 2 materials-12-02331-f002:**
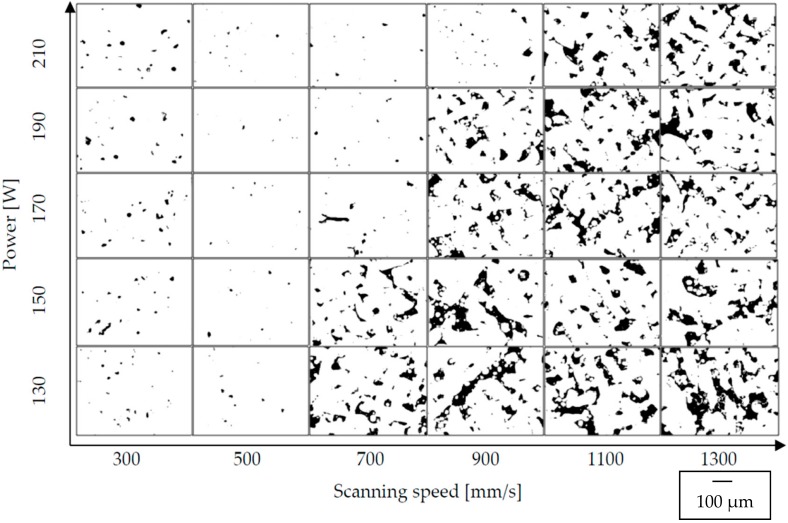
Graphical representation of the variable pore morphology in vertical samples manufactured with Direct Metal Laser Sintering (DMLS) technology, perpendicular to the build direction; 100×.

**Figure 3 materials-12-02331-f003:**
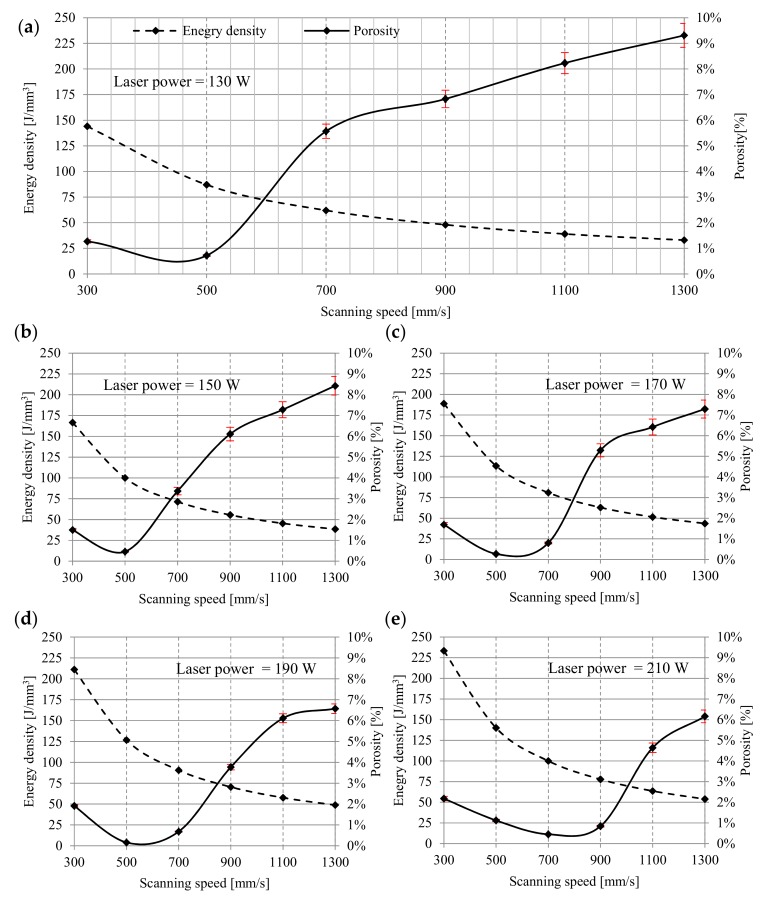
The influence of scanning speed and energy density on the porosity: (**a**) 130 W, (**b**) 150 W, (**c**) 170 W, (**d**) 190 W, (**e**) 210 W.

**Figure 4 materials-12-02331-f004:**
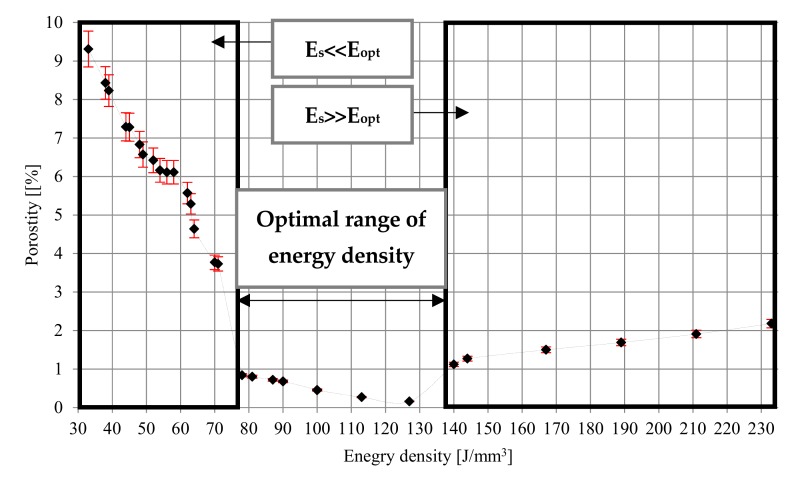
Relation of porosity and energy density for Ti6Al4V DMLS alloy.

**Figure 5 materials-12-02331-f005:**
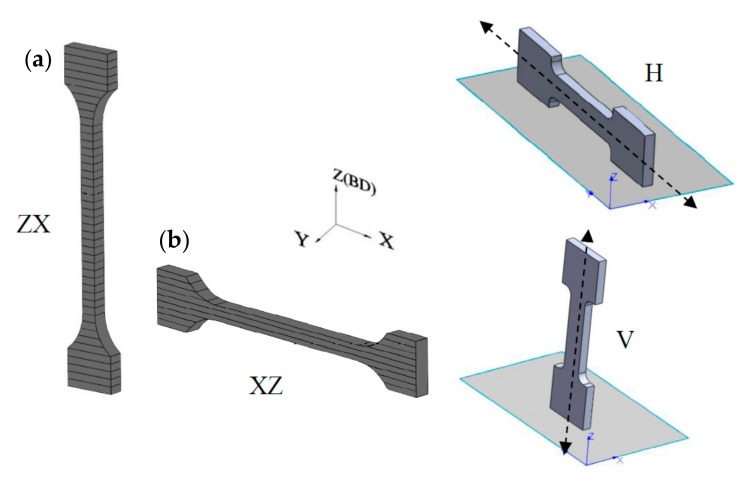
Tensile samples oriented in: (**a**) XZ plane and (**b**) ZX plane.

**Figure 6 materials-12-02331-f006:**
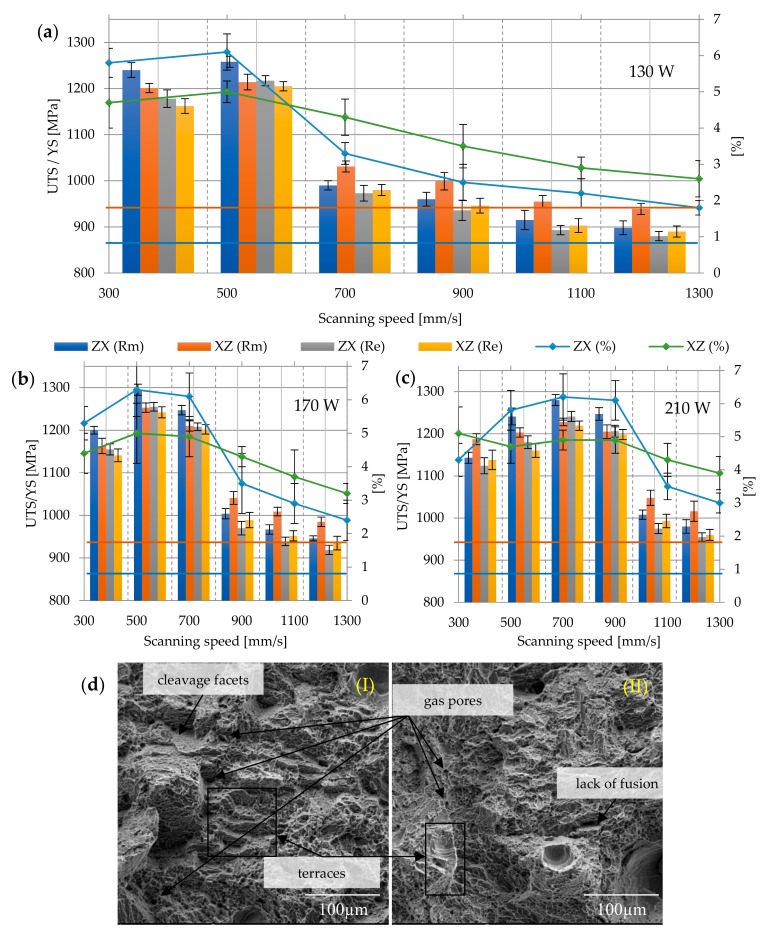
Strength parameters of as–built samples with different laser speeds and powers: (**a**) 130 W, (**b**) 170 W, (**c**) 210 W, and (**d**) fracture surface: (I) 130 W, 300 mm/s, ZX; (II) 130 W, 300 mm/s, XZ; 900×; Rm min 
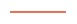
 and Re min 
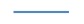
 for Ti6Al4V in accordance with ASTM F 1472.

**Figure 7 materials-12-02331-f007:**
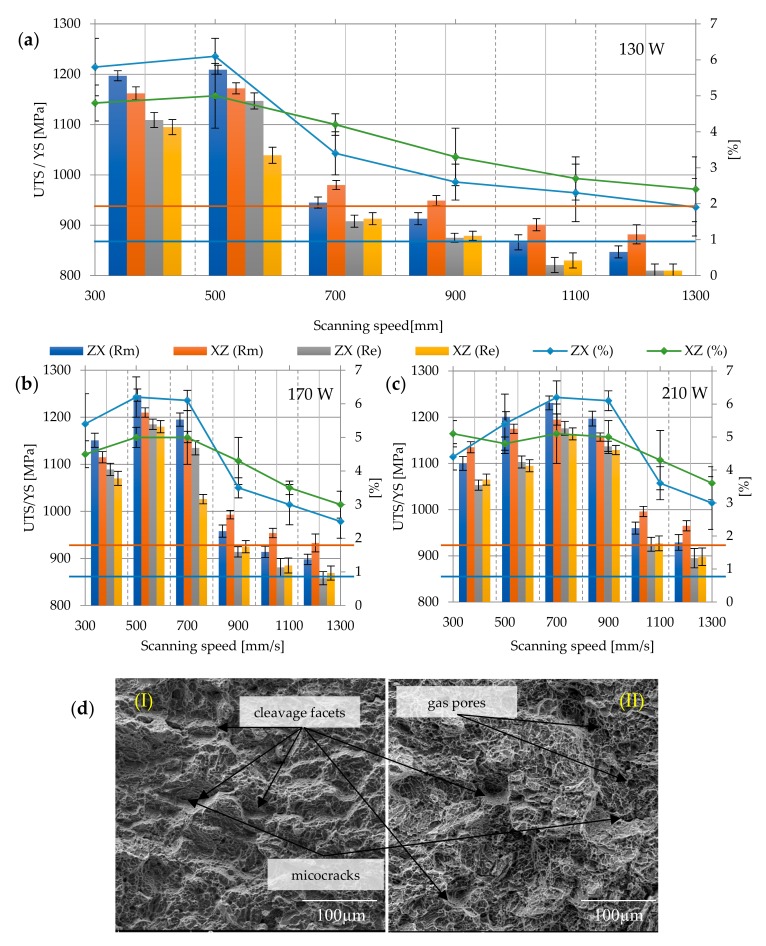
Strength parameters of samples heat treated at 650 °C with different laser speeds and powers: (**a**) 130 W, (**b**) 170 W, (**c**) 210 W, and (**d**) fracture surface: (I) 130 W, 300 mm/s, ZX; (II) 130 W, 300 mm/s, XZ; 900×; Rm min 
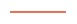
 and Re min 
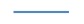
 for Ti6Al4V in accordance with ASTM F 1472.

**Figure 8 materials-12-02331-f008:**
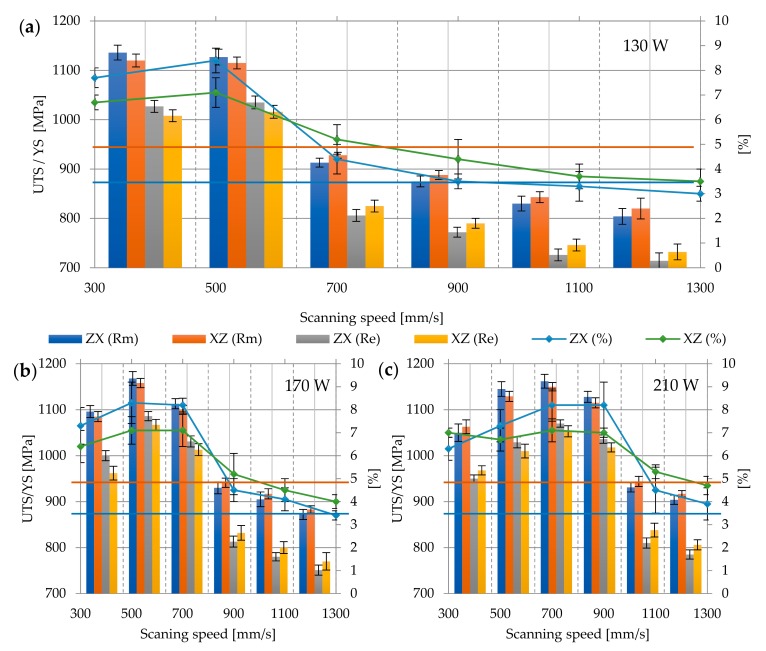
Strength parameters of samples heat treated at 750 °C with different laser speeds and powers: (**a**) 130 W, (**b**) 170 W, (**c**) 210 W, and (**d**) fracture surface: (I) 130 W, 300 mm/s, ZX; (II) 130 W, 300 mm/s, XZ; 900×; Rm min 
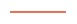
 and Re min 
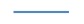
 for Ti6Al4V in accordance with ASTM F 1472.

**Figure 9 materials-12-02331-f009:**
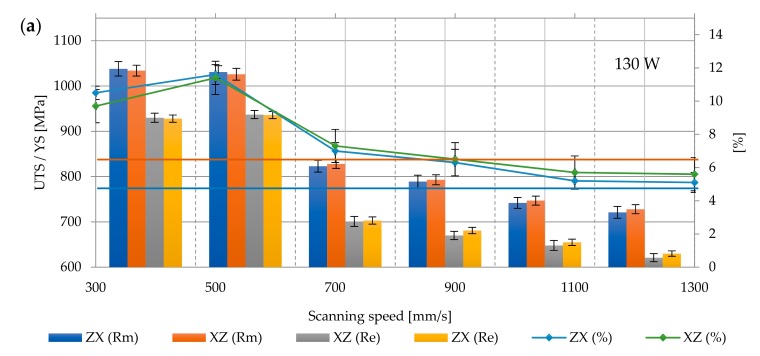
Strength parameters of samples heat treated at 850 °C with different laser speeds and powers: (**a**) 130 W, (**b**) 170 W, (**c**) 210 W, and (**d**) fracture surface: (I) 130 W, 300 mm/s, ZX; (II) 130 W, 300 mm/s, XZ; 900×; Rm min 
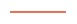
 and Re min 
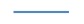
 for Ti6Al4V in accordance with ASTM F 1472.

**Figure 10 materials-12-02331-f010:**
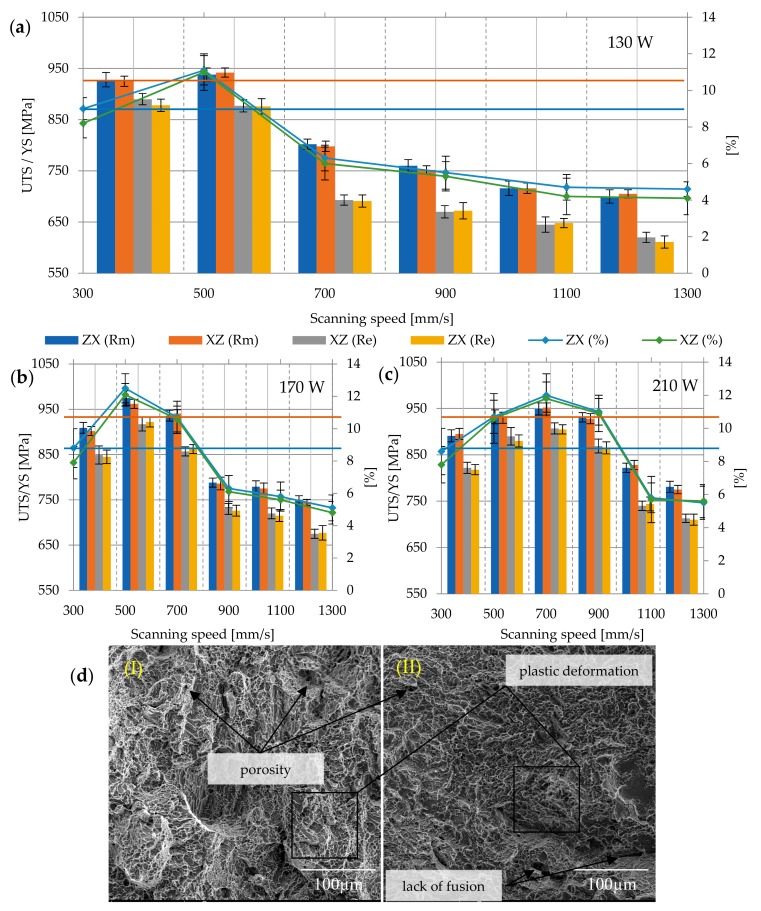
Strength parameters of samples heat treated at 650 °C with different laser speeds and powers: (**a**) 130W, (**b**) 170W, (**c**) 210W, and (**d**) fracture surface: (I) 130 W, 300 mm/s, ZX; (II) 130 W, 300 mm/s, XZ; 900×; Rm min 
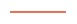
 and Re min 
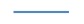
 for Ti6Al4V in accordance with ASTM F 1472.

**Figure 11 materials-12-02331-f011:**
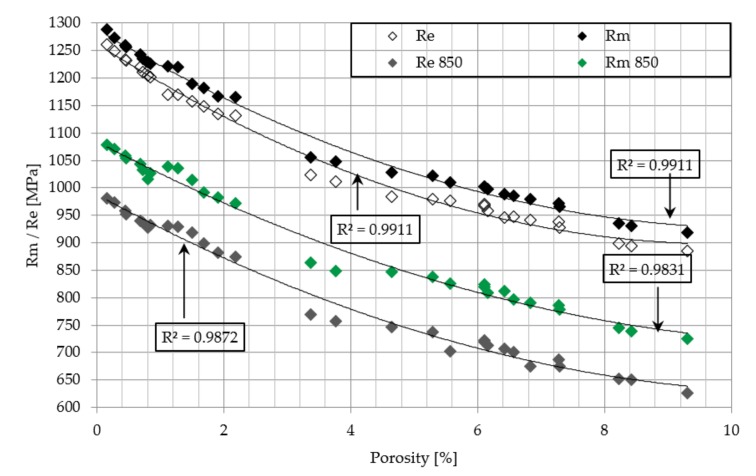
Influence of porosity on Ultimate Tensile Strength and Yield Strength of as-build and heat treated at 850 °C samples.

**Figure 12 materials-12-02331-f012:**
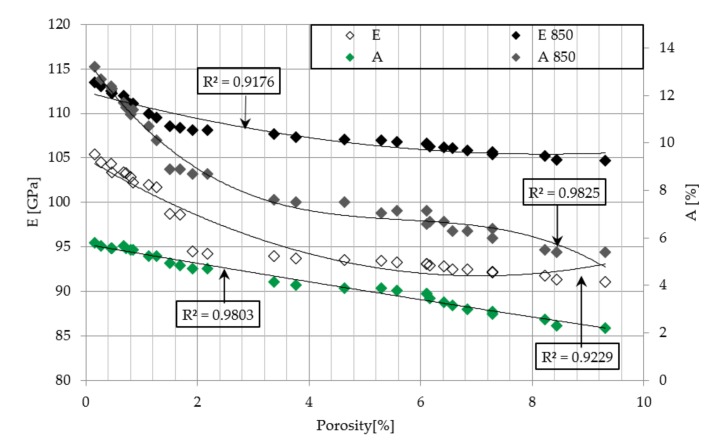
Influence of porosity on Young modulus and elongation of as-build and heat treated at 850 °C samples.

**Figure 13 materials-12-02331-f013:**
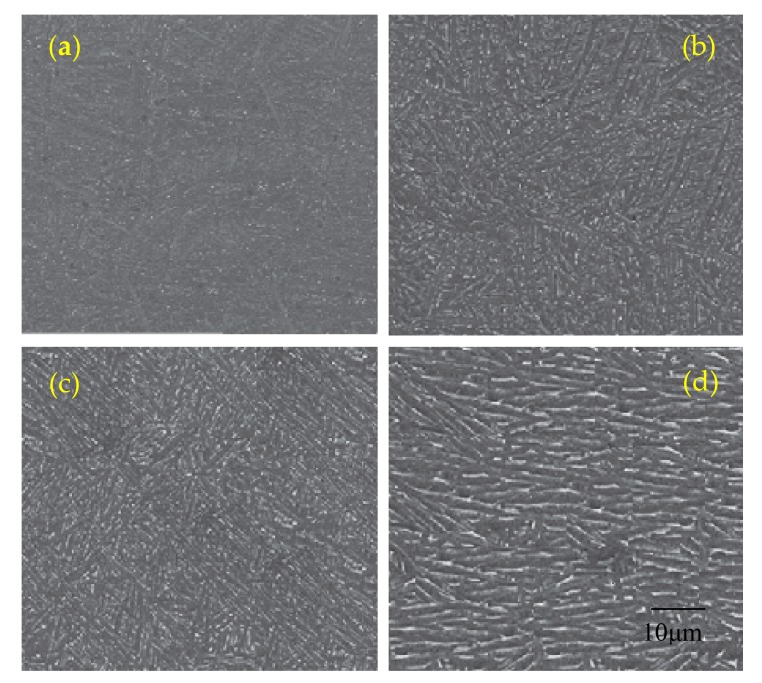
SEM images of microstructure of heat treated samples: (**a**) 650 °C, (**b**) 750 °C, (**c**) 850 °C, and (**d**) 950 °C.

**Figure 14 materials-12-02331-f014:**
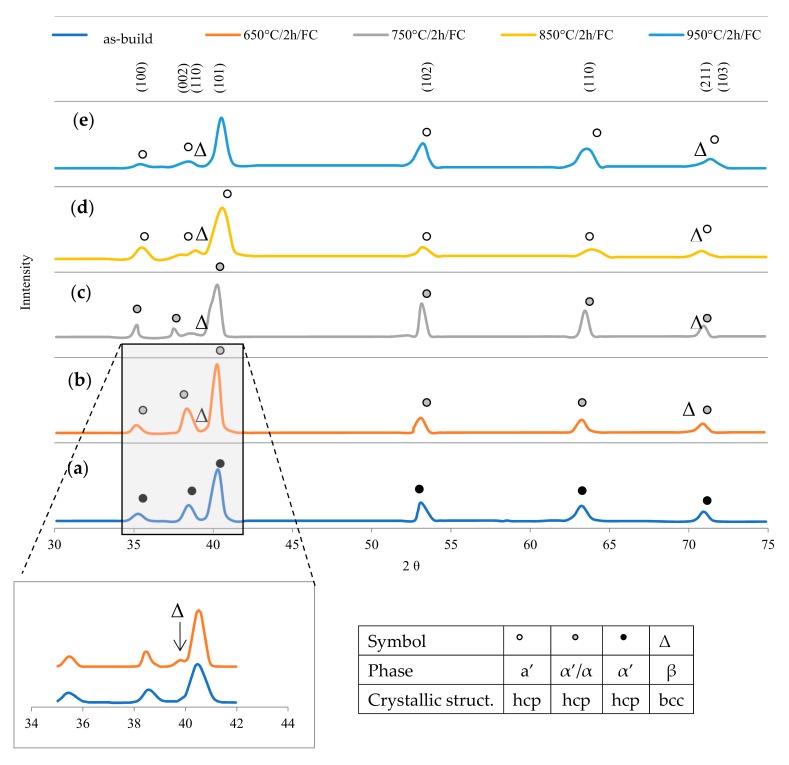
XRD spectrum of Ti6Al4V alloy DMLS: (**a**) as-build, (**b**) annealing at 650 °C, (**c**) annealed at 750 °C, (**d**) 850 °C, and (**e**) 950 °C for 2 h, cooled with a furnace.

**Table 1 materials-12-02331-t001:** Chemical composition and material characteristics of used powder.

Element	Al	V	O	N	H	Fe	C	Ti
wt.%	5.97	4.04	0.195	0.036	0.010	0.24	0.061	Bal.
ASTM F 2924-14	5.5–6.75	3.5–4.5	<0.2	<0.05	<0.015	<0.3	<0.08	Bal.

**Table 2 materials-12-02331-t002:** Processing parameters.

**Scanning Velocity (mm/s)**	**Laser Power [W]**				
	130	150	170	190	210
	**Energy Density (J/mm^3^)**				
300	144	166	188	211	233
500	87	100	113	127	140
700	62	71	81	90	100
900	48	56	63	70	74
1100	40	45	52	58	64
1300	33	38	44	49	54

**Table 3 materials-12-02331-t003:** Dimensional accuracy of samples for testing mechanical properties.

Dimension	Thickness A–E [mm]	Necking H/I/J [mm]	Width F/G/K/L [mm]	Length M–N [mm]
Nominal	2	2	7	45
Min	2.48	2.07	6.97	45.55
Max	2.55	2.17	7.16	45.65
Average	2.51	2.12	7.07	45.58
Standard deviation	0.01	0.02	0.06	0.03

**Table 4 materials-12-02331-t004:** List of DMLS parameters and properties of Ti6Al4V alloy heat treated at 850 °C meeting the criteria of ASTM F 1142 standard for implantable biomaterials.

P [W]	V [mm/s]	Rm [MPa]		Re [MPa]		A [%]		E [GPa]		Es [J/mm^3^]	P [%]
		ZX	XZ	ZX	XZ	ZX	XZ	ZX	XZ		
150	500	1056 ± 13	1052 ± 12	954 ± 10	950 ± 8	13.4 ± 0.5	13.2 ± 0.6	106.5 ± 1.3	106.2 ± 0.6	100	0.46
170	500	1074 ± 13	1069 ±10	974 ± 9	973 ± 7	13.7 ± 0.6	13.6 ± 0.5	107.5 ± 0.7	107.5 ± 0.8	113	0.27
190	500	1080 ± 10	1077 ± 11	982 ± 9	980 ± 9	14.3 ± 06	14.1 ± 0.7	108.0 ± 1.2	108.7 ± 0.5	127	0.16
210	700	1062 ± 12	1055 ± 10	962 ± 9	954 ± 8	13.5 ± 0.5	13.3 ± 0.8	107.2 ± 1.0	107.4 ± 0.9	100	0.45

**Table 5 materials-12-02331-t005:** Change the grain size in respect of energy density and temperature.

Heat Treatment	Energy Density [J/mm^3^]			
	33	100	233	Average
AB	0.17 ± 0.05	0.25 ± 0.08	0.38 ± 0.06	0.27 ± 0.11
650 °C	0.42 ± 0.11	0.57 ± 0.12	0.81 ± 0.11	0.60 ± 0.22
750 °C	0.81 ± 0.10	1.16 ± 0.15	1.63 ± 0.09	1.20 ± 0.41
850 °C	1.17 ± 0.12	1.49 ± 0.13	2.11 ± 0.11	1.59 ± 0.48
950 °C	1.83 ± 0.15	2.31 ± 0.18	2.80 ± 0.21	2.31 ± 0.49

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
