# Peer review of "Effect of Laser Energy Density, Internal Porosity and Heat Treatment on Mechanical Behavior of Biomedical Ti6Al4V Alloy Obtained with DMLS Technology"

_materials, 2019, doi:10.3390/ma12142331_

Reviewer 1 Report

This work investigated the fffect of laser energy density, internal porosity and heat treatment on mechanical behavior of biomedical Ti6Al4V alloy obtained by selective laser melting. Although the novelty is not high, the reviewer still recommends to accept this manuscript because this work has presented some useful basic data. However, revision is required by addressing the following comments.

1. Proof reading is required to remove errors and typos.

2. All abbreviations should be fully described in the first use.

3. Abstract is too long. Please make the abstract concise and contain more results.

4. Figure 2 should include scale bars.

5. Figure 3 and Figure 4 should contain error bars. The change in density with laser power/scanning speed has been widely reported in different titanium alloys, such as Ti-24Nb-4Zr-8Sn, Ti-6lAl-4V and CP-Ti. Some review papers on Selective Laser Melting of Titanium Alloys and Titanium Matrix Composites for Biomedical Applications have also mentioned the optimal laser energy density of 120 J/mm3 to produce high density samples. These papers should be cited and compared.

6. It is not recommended too many similar figures of Figure 6-10 in the manuscript. The mechanical properties data have been summarized in Table 5. Therefore, please remove most Figures from Figure 6-10.

7. The reasons for the resultant mechanical properties should be discussed thoroughly based on the microstructure. In addition, it is also suggested to add some discussion (by citing papers) on the change of phase (alpha prime to alpha) for corrosion behaviors of Ti-6Al-4V.

Author Response

Dear Sir / Madam
I would like to thank You very much for the comments I received. I am a Young scientist, therefore any advice given to me will certainly improve my research skills and the ability to describe my research.

Based on the comments received from three reviewers, the abstract was rearranged and I presented more results.

I completed  Figure 2 with a scale bars, and Figure 3 and 4 with an error bars.

I also pointed out several other articles in which similar studies were carried out and the results obtained in them, according to Your instructions.

Indeed, individual figures 6-10 did not differ significantly. My goal, however, was to show how a change in the DLMS paramters affects the change of UTS, YS and A parameters in each sample. However, with the recommendation suggested, some of the charts were omitted.

Changes in the microstructure are described in sub-chapter 2.3. In addition, at the request of another reviewer, microscope images showing microstructures and analysis of grain size changes after heat treatment were added.

I would also like to thank You for the consideration of the need to analyze micro-corrugated board for its corrosion resistance. I am in the process of preparing the next article, in which I will present the results of the corrosion resistance of samples, samples of cytotoxicity and bioactivity tests, depending on the quality of the surface. Therefore, because in this article it does not contain any results related to corrosion resistance, I allowed myself not to include in it a discussion about this issue.

I very much hope that the amendments that I have applied are satisfactory and will allow You to give a positive opinion on the article.
With heartfelt thanks,

Żaneta Anna Mierzejewska

Reviewer 2 Report

The author has evaluated the effect of the process parameters and different heat treatments on the mechanical properties of the Ti64 alloy. Although this topic has been studied before in several articles, however, a very wide range of parameters have been considered in this study that makes the work quite useful and interesting. The English language of the text should be significantly improved, the author may ask a native speaker for this aim. The structure of the manuscript should be modified. The current structure is not easy to follow and has some flaws. The manuscript can be considered for the second revision after MAJOR REVISION.

- Line 13: delete “on” in the statement “Significantly affect on porosity …”

- Line 13: replace “morfology” with the English form “morphology”

- What exactly does the author mean by “breaking resistance”? this is not a common term to be used, please replace it with a more meaningful and specific word.

- Lines 17 and 18: the grammar of the sentence is wrong, using two verbs wrongly in one statement. Probably, the verb “suggests” can be replaced by “suggestion”.

- By “columnar beta grains”, do you mean “columnar PRIOR beta grains”?

- The literature review about the effect of porosity on static and fatigue behavior of additively manufactured Ti6Al4V material can be improved. There is almost no significant literature review on this important aspect of your research in your manuscript. I recommend the author to use the following researches to improve this point:

*Effect of microstructure and defects on fatigue behaviour of directed energy deposited Ti–6Al–4V. Sci Technol Weld Joi 2015; 20(8): 659-669.

**Porosity effect on tensile behavior of Ti-6Al-4V specimens produced by laser engineered net shaping, Proceedings of the Institution of Mechanical Engineers, Part C: Journal of Mechanical Engineering Science. (in-press) (DOI: 10.1177/0954406218813384)

***Fatigue behavior of porous Ti-6Al-4V made by Laser Engineered Net Shaping, Materials, 11(2) (2018) 284. (DOI: 10.3390/ma11020284)

- Line 106: in the whole text, after the word “different”, the plural form of the noun and plural verb should be used. “Different beam speeds were used.” (speeds and not speed, were and not was). Please correct this in the whole text.

- Please provide the nominal dimensions of the test specimens in a separate figure. The length of the middle (reduced) part and the radius of the notched region should be indicated.

- the section “results and discussions” is very long without any subsection. It is very difficult to follow the text in this way. I recommend the author to divide this section to some subsection and connecting the discussions in those subsections at the end of the section.

- Lines 240, 267, 286, and 303: this way of naming the subsections only by a number (temperature) is absolutely not proper for a scientific work. Please modify these titles.

- The number of sections is wrong! The naming is also wrong. There is section 3 with the title “results and discussion” this means that you have both results and the discussions in this section. However, the next section that should be section 4 doesn’t exist in the text. Section 5 is again called “discussions”! didn’t you provide the discussions in section 3? Please add section 4 (which is missing) and either merge section 5 with 3 or correct the title of section 3.

- since the manuscript is written by a single author, the author contribution section wouldn’t seem necessary, especially in this long form that has been provided in the text.

- the author has stated the effect of the heat treatment on the grain size in the text. Can you please add a table providing the grain size for different heat treatment conditions? (and possibly for different process parameters)

- Can the author provide some limited information about the failure mechanisms (using SEM analysis) in different cases of printing and also different heat treatment conditions?

Author Response

Response to Reviewer 2 Comments

Dear Sir / Madam
I would like to thank You very much for the comments I received. I am a Young scientist, therefore any advice given to me will certainly improve my research skills and the ability to describe my research.

Based on the comments received from three reviewers, I rearranged article.

I very much hope that the amendments that I have applied are satisfactory and will allow You to give a positive opinion on the article.

With heartfelt thanks,

Żaneta Anna Mierzejewska

Point 1 - 5:

- Line 13: delete “on” in the statement “Significantly affect on porosity …”

- Line 13: replace “morfology” with the English form “morphology”

- What exactly does the author mean by “breaking resistance”? this is not a common term to be used, please replace it with a more meaningful and specific word.

- Lines 17 and 18: the grammar of the sentence is wrong, using two verbs wrongly in one statement. Probably, the verb “suggests” can be replaced by “suggestion”.

- By “columnar beta grains”, do you mean “columnar PRIOR beta grains”?

Response 1 - 5:

The mistake has been corrected in the manuscript.

Point 6: The literature review about the effect of porosity on static and fatigue behavior of additively manufactured Ti6Al4V material can be improved. There is almost no significant literature review on this important aspect of your research in your manuscript (...).

Response 6:

As suggested, this part of the article was extended to discuss the influence of porosity on the mechanical properties of samples obtained with DMLS technology.

Point 7: Line 106: in the whole text, after the word “different”, the plural form of the noun and plural verb should be used. “Different beam speeds were used.” (speeds and not speed, were and not was). Please correct this in the whole text.

Response 7: The mistake has been corrected in the manuscript.

Point 8 - 12: - Please provide the nominal dimensions of the test specimens in a separate figure. The length of the middle (reduced) part and the radius of the notched region should be indicated.

- the section “results and discussions” is very long without any subsection. It is very difficult to follow the text in this way. I recommend the author to divide this section to some subsection and connecting the discussions in those subsections at the end of the section.

- Lines 240, 267, 286, and 303: this way of naming the subsections only by a number (temperature) is absolutely not proper for a scientific work. Please modify these titles.

- The number of sections is wrong! The naming is also wrong. There is section 3 with the title “results and discussion” this means that you have both results and the discussions in this section. However, the next section that should be section 4 doesn’t exist in the text. Section 5 is again called “discussions”! didn’t you provide the discussions in section 3? Please add section 4 (which is missing) and either merge section 5 with 3 or correct the title of section 3.

- Since the manuscript is written by a single author, the author contribution section wouldn’t seem necessary, especially in this long form that has been provided in the text

Response 8 -12: Thank You for paying attention to these mistakes, they have already been corrected

Point 9 - 10: - the author has stated the effect of the heat treatment on the grain size in the text. Can you please add a table providing the grain size for different heat treatment conditions? (and possibly for different process parameters)

- Can the author provide some limited information about the failure mechanisms (using SEM analysis) in different cases of printing and also different heat treatment conditions?

Response 9 - 10: The article was supplemented with a table with the grain size as a function of changes in energy density and heat treatment temperature as well as SEM images of fracture surface.

Reviewer 3 Report

Please find the attached report.

Author Response

Response to Reviewer 2 Comments

Dear Sir / Madam
I would like to thank You very much for the comments I received. I am a Young scientist, therefore any advice given to me will certainly improve my research skills and the ability to describe my research.

Based on the comments received from three reviewers, I rearranged article.

The fact is that there are many articles about the impact of the parameters I examine on the mechanical strength and porosity of the material. However, most of them concern selected and well-defined parameters. In my work, the changes in mechanical properties caused by the change of energy density for a wider range of variables were investigated.

In accordance with the guidelines, the literature part of the work has been re-arranged.
It presents the results of other scientists' research on mechanical properties and microstructure obtained in SLM / DMLS processes, as well as properties of Ti6Al4V

According to the guidelines, the literature has been changed. Presents the results of other researches on mechanical properties and microstructure obtained in SLM / DMLS processes, as well as properties of Ti6Al4V indicating its suitability in biomedical engineering

The porosity of the samples was tested in the entire volume of the sample using a microtomorgaphy. Therefore, the porosity for samples melted in both vertical and horizontal orientations was not investigated separately. Thus, the porosity shown in diagram 3 is the average value of the total porosity tested for the five random samples.

Due to the limited amount of space, the legend to the charts it's under the first one. Vertical samples have been described as ZX, horizontal samples as XZ. This information was included in the revised abstract and sub-chapter 2.2.

The XRD study for random samples did not indicate differences in the phase composition, and the generated graphs were almost no different from each other. The graph presented in the paper applies to samples with an energy density of 100 J/mm3 and melted in vertical orientation, which was mentioned at work.

I very much hope that the amendments that I have applied are satisfactory and will allow You to give a positive opinion on the article.

With heartfelt thanks,

Żaneta Anna Mierzejewska

Round  2

Reviewer 1 Report

Thanks for the revision. However, it is hard to read the manuscript. After reading through the revision, it is found that the two key concerns in previous review were not replied. the reviewer is not satisfied with the revision.

1. Figure 3 and Figure 4 should contain error bars. The change in density with laser power/scanning speed has been widely reported in different titanium alloys, such as Ti-24Nb-4Zr-8Sn, Ti-6lAl-4V and CP-Ti. Some review papers on Selective Laser Melting of Titanium Alloys and Titanium Matrix Composites for Biomedical Applications have also mentioned the optimal laser energy density of 120 J/mm3 to produce high density samples. These papers should be cited and compared.

2. The reasons for the resultant mechanical properties should be discussed thoroughly based on the microstructure. In addition, it is also suggested to add some discussion (by citing papers) on the change of phase (alpha prime to alpha) for corrosion behaviors of Ti-6Al-4V.

Young scientist is not an excuse to provide insufficient quality manuscript.

Author Response

Dear Sir / Madam

I have completed  Figure 3 and 4 with an error bars.

I also pointed out several other articles in which similar studies were carried out and the results obtained in them, in results and discussion section, directly under Fig. 4.

I also complemented the text with the discussion related to the change of the microstructure and its resistance to corrosion phenomena.

I hope that in this form the article is acceptable.

With heartfelt thanks,

Żaneta Anna Mierzejewska

Reviewer 2 Report

Most of the comments have been responded by the author. Minor suggestion: If possible, replace the SEM pictures with higher quality pictures and also add a clear scale bar. The location of pores can be also indicated on the picture. The manuscript can be accepted for publication after this.

Author Response

Dear Sir / Madam
SEM photographs have high resolution, but it seems to me that the quality of the pictures has been degraded by converting the file to pdf.

I will ask the editor for other conversion settings for these photos.

Thank you kindly for any tips and comments, thanks to which the article has been corrected.

I very much hope that the amendments that I have applied are satisfactory and will allow You to give a positive opinion on the article.

With heartfelt thanks,

Żaneta Anna Mierzejewska

Reviewer 3 Report

Please, find the attached report.

Author Response

Response to Reviewer 3

Dear Sir / Madam

Thank you very much for the insightful review.

 I would like to refer more fully to Notes 4 and 5. Indeed, the final effect of the DMLS process is influenced by: laser power, scanning speed, hatch spacing and layer thickness. All of these parameters are taken into account when calculating the energy density, based on a 1 (p. 4). The change of any of these parameters affects the change in energy density. In my research, I included two of the four variables. Referring to Note 5 - the quality of the images is very much lost when converting to pdf. Fig. 2 is a binary representation of the microstructure, hence black pixels mean pores and white - solid surroundings. In the file that I send in word format to the journal, the quality is definitely better.

Indeed, it is possible to place all of the data in Fig. 3 in two diagrams, but these data become illegible and difficult to read.

Taking into account the comments of your and other two reviewers, I have improved the manuscript and I hope that it is acceptable in this form.
With heartfelt thanks,

Żaneta Anna Mierzejewska

Round  3

Reviewer 1 Report

The revision is acceptable for publication

Author Response

Dear Sir / Madam

I would like to thank you once again for all the comments and advices regarding the manuscript that influenced the quality of my work.
Yours sincerely,

Żaneta Anna Mierzejewska

Reviewer 3 Report

Please find the attached report.

Author Response

(The authors gave the same response as above.)
